# The effect of peer relationships on college students' behavioral intentions to be physically active: The chain-mediated role of social support and exercise self-efficacy

Xielin Zhou[1], Mu Zhang[1,2]*, Lu Chen[1], Bo Li[1], Jiao Xu[3]

1 Department of Sports Training, Chengdu Sport University, Chengdu, Sichuan, China, 2 Information Technology Centre, Chengdu Sport University, Chengdu, Sichuan, China, 3 School of Physical Education, Sichuan Technology and Business University, Meishan, Sichuan, China

☯ These authors contributed equally to this work.
* zhangmucdsuedu@163.com

## Abstract

### Purpose

Physical activity behavioral intentions have a positive impact on individuals' physical and mental health, social adjustment, and academic work. This study explored the effects of peer relationships on college students' behavioral intentions for physical activity and analyzed the mediating role of social support and exercise self-efficacy.

### Methods

The Peer Relationship Inventory, the Physical Activity Rating Scale-Intentions to Physical Activity Behavior subscale, the Social Support Scale, and the Exercise Self-Efficacy Scale were used to psychometrically measure 514 college students (age = 19.00 ± 1.27) from June 3, 2024, to June 7, 2024, and the Bootstrap method was used to mediate the relationship between social support and exercise self-efficacy.

### Results

(1) Peer relationships, social support, exercise self-efficacy and intention to engage in physical activity behaviors were positively correlated (all p < 0.01); (2) gender differences in exercise self-efficacy and intention to engage in physical activity behaviors existed (p < 0.001), and male students scored higher than female students (M ± SD: 3.18 ± 0.55 > 2.98 ± 0.59; 3.23 ± 0.62 > 2.95 ± 0.61); (3) the mediating effect size of social support (peer relationship→social support→behavioral intention to exercise) was 0.078, which accounted for 8.83% of the total effect, and that of exercise self-efficacy (peer relationship→exercise self-efficacy→behavioral intention to exercise) was 0.314, which accounted for 35.56% of the total effect, the chained mediation

**Data availability statement:** All relevant data are within the manuscript and its Supporting Information files.

**Funding:** The author(s) received no specific funding for this work.

**Competing interests:** The authors have declared that no competing interests exist.

effect of social support and exercise The chain mediation effect size of self-efficacy (peer relationship→social support→exercise self-efficacy→behavioral intention to exercise) was 0.057, accounting for 6.46% of the total effect, and none of the confidence intervals contained 0.

## Conclusions

(1) College students' peer relationships can significantly and positively predict behavioral intention to exercise; (2) college students' peer relationships can have a simple influence on the behavioral intention to exercise through social support and exercise self-efficacy, respectively; (3) College student peer relationships can also influence intention to engage in physical activity through social support and exercise self-efficacy in a chained manner.

## 1. Introduction

Behavioral intention to engage in physical activity is an antecedent to individual participation in physical activity and is an important factor affecting the overall development of college students' bodies and minds [1]. In the face of a complex social environment and severe employment pressure, how to promote their physical and mental health development has become a key concern of the state [2,3]. However, at present, some students' health awareness is weak, irregular work and rest, and many health-damaging lifestyles have become hidden dangers that threaten students' health [4]. According to the survey data of China Youth Net Campus News Agency, the proportion of college students who exercise less than three times a week is as high as 48.19%, and 58.7% of college students exercise for less than 30 minutes each time [5]. The Opinions on Comprehensively Strengthening and Improving School Physical Education Work in the New Era states, "Strengthening the positive role of school physical education work in the new era in promoting students' quality education and all-round development" [6]; and the Special Action Plan for Comprehensively Strengthening and Improving Students' Mental Health Work in the New Era, jointly issued by the Ministry of Education and other seventeen departments, proposes that. "Adhere to the concept of health-first education, effectively place mental health work in a more prominent position, and promote the coordinated development of students' ideological and moral qualities, scientific and cultural qualities, and physical and mental health qualities [7]." The focus of physical education in China has gradually shifted to the development of student's physical and mental health, and the active promotion of college students' commitment to physical exercise and the development of lifelong sports habits has become the core work and important issue in the field of physical education in schools in China.

Physical activity is an indispensable part of people's lives, as it reflects their sporting values, physical fitness, and mental health [8]. Analyzing from a doctrinal perspective, existing studies have mainly explored the relationships between peer relationships and physical activity behavior [9], social support and physical activity behavior [10], and exercise self-efficacy and physical activity behavior [11]. Some studies have found that good peer relationships (i.e., positive peer interactions through emotional support, joint

sports participation, and healthy competition) can promote individuals' engagement in physical activity, which is one of the most important motivations for college students' physical and mental health development [12]. Specifically, under the leadership of peers, individuals will generate behavioral intentions to engage in physical activity through social modeling (e.g., observing peers' exercise behaviors), social facilitation (e.g., peers' presence to enhance exercise performance), and attributional need satisfaction (e.g., integrating into the exercise community), thus participating in sport more actively [13]. Although many previous studies have explored the influence mechanism of physical activity and achieved certain results, few scholars have explored the influence mechanism of the antecedent factor of physical activity behavior - the intention of physical activity behavior. So, what is the influence mechanism of physical activity behavior intention? Is it more influenced by intrinsic or extrinsic influences? According to existing research, the formation of the intention to engage in physical activity is a complex psychological process that is influenced by a combination of factors. Bandura's triadic interaction theory suggests that environmental factors (e.g., peer relationships, and social support) form a cyclic dynamic with behavioral intention through personal cognition (e.g., self-efficacy). Based on the ternary interaction theory, this study introduces peer relationship and social support among environmental factors and exercise self-efficacy among personal cognitive factors to explore the influence mechanism of physical activity behavioral intention, aiming at enriching the research results in the existing field, and providing theoretical support for the development of physical education and lifelong physical education in colleges and universities.

## 2. Rationale and assumptions

### 2.1. The effect of peer relationships on college students' behavioral intentions to be physically active

Peer relationships are defined as interpersonal relationships between people of the same age or similar levels of psychological development who engage in common activities or collaborate. These relationships are considered to be an important influence on the socialization of individuals as they grow [14]. The establishment of positive peer relationships has been demonstrated to provide university students with the necessary emotional support, thereby reducing the stress of academic and professional obligations. Furthermore, such relationships can also promote participation in individual sports [15]. The notion of behavioral intention to physical activity can be defined as an antecedent to an individual's participation in said activity. This concept refers to an individual's inclination or intention to intentionally engage in physical activity, reflecting an individual's positive attitude and readiness to engage in physical activity. The influence of this concept is derived from a variety of aspects, including personal motivation, the exercise environment, and peer relationships [16–18]. It has been suggested that peer relationships are an important predictor of physical activity behavior [19]. At the same time, in college physical education and health courses, students further improve their cooperation ability through mutual collaboration in course activities and establishing good peer relationships, which can make them more willing to participate in physical exercise. Thus, peer relationships have an important impact on college students' physical fitness and health. In the current research, most of the studies focus on the influence of peer relationships and physical activity behavior, but there are fewer analyses and quantitative studies on the intrinsic mechanism of the intention of peer relationships and physical activity behavior. The Theory of Planned Behavior suggests that an individual's behavior is influenced by his or her behavioral intentions, which in turn are influenced by behavioral attitudes, subjective norms, and perceived behavioral control [20]. Through this theory, peer relationships may influence individuals' behavioral attitudes and perceptual behavioral control by influencing their behavioral intentions, so what is the effect of peer relationships on physical activity behavioral intentions in college student groups? Based on this, this study proposes Hypothesis H1: Peer relationships can significantly and positively predict physical activity behavioral intention.

### 2.2. The mediating role of social support in peer relationships and physical activity behavioral intentions

Social support is an interactive relationship in which individuals give each other emotional or material support, which plays an important role in the development of a person's physical and mental health [21]. The Theory of Planned Behavior suggests that a person's behavioral intentions are influenced by attitudes, subjective norms, and perceptual control

[22]. When college students are supported by social groups (e.g., parents, classmates, and friends) in the process of generating behavioral intentions, the probability that they will put their intentions into action increases. It has been shown that social support positively predicts physical activity behavioral intentions [23], and that individuals generate a positive emotional value when they feel support from others for their participation in physical activity behaviors, which reinforces their physical activity behavioral intention and allows them to put this behavioral intention into practice. Meanwhile, self-determination theory divides people's motivation into two parts: internal motivation (originating from the individual himself) and external motivation (originating from the stimulation of external factors), and social support as an external motivation can promote individuals' intention of physical exercise behavior by satisfying their basic psychological needs [24], and scholars have further obtained the results of the empirical research from the viewpoint of self-determination theory that the effect of social support on the exercise intention of college students is significant. College students' exercise intention is significant [25].

Related studies have also found that peer relationships are significantly positively correlated with social support, and the more positive peer relationships college students have, the more support they receive from their peers [26]. On the one hand, peer relationships are an important source of social support in university study and life, and when individuals establish good peer relationships with their peers, they can obtain emotional support, understanding, and action acceptance through communication with their peers [27]. On the other hand, from the perspective of attachment theory, when college students participate in school-related activities, the attachment relationship formed between them and their peers will have an important impact on their emotions and behaviors [28]. For example, when college students face difficulties and frustrations in sports activities, positive peer relationships can give them emotional and operational support through encouragement and enhance their confidence in coping with difficulties. Based on this, peer relationships can positively predict social support, and social support in turn can influence physical activity behavioral intention. Therefore, this study proposes the hypothesis H2: social support plays a mediating role in peer relationships and physical activity behavior intention.

## 2.3.  The mediating role of exercise self-efficacy in peer relationships and physical activity behavioral intentions

Exercise self-efficacy is the cognitive ability of an individual to believe that he/she is able to accomplish the set goals and tasks of physical activity, and the higher the exercise self-efficacy of an individual, the more he/she is able to manage and regulate their behavior [29]. As one of the predictors of individual behavioral change, exercise self-efficacy can influence cognitive processes, which in turn have a significant impact on behavioral selectivity, persistence, and the acquisition of new behaviors as well as the performance of acquired behaviors [30], from the perspective of self-efficacy theory, when individuals have a higher sense of exercise self-efficacy, the stronger their intention to engage in physical activity behaviors, which in turn lead to the formation of healthy behavioral habits. thereby forming healthy behavioral habits [31]. It has been found that self-efficacy and physical activity have a close relationship and are stable throughout all stages of physical activity, including the intention generation stage and the action stage [11], In other words, exercise self-efficacy can significantly influence individuals' intention to engage in physical activity. Scholars have further pointed out that self-efficacy can determine the selectivity and persistence of behaviors by optimizing individuals' cognitive processes [32]. Thus, exercise self-efficacy can positively predict physical activity behavioral intention.

Self-efficacy can mediate between peer relationships and certain other variables, which include internet addiction, mental health, and life satisfaction [33,34]. So can exercise self-efficacy mediate between peer relationships and physical activity behavioral intentions, and is this mediation partially or fully mediated? The answer to this question needs to be further explored. College students' self-efficacy is influenced by factors such as successful experiences, alternative experiences, and extrinsic emotional acquisition [35]. Previous studies have shown that peers are important role models in college students' learning and life, and their excellent behavior and inner cultivation positively influence individuals and enhance self-efficacy through imitation and learning [36]. From this perspective, in a university setting, individuals

cooperate with their peers to accomplish goals and tasks related to physical activity, which enhances peer relationships while increasing both parties' exercise self-efficacy. Therefore, based on the above elaboration, this study proposes Hypothesis H3: Exercise self-efficacy plays a mediating role in peer relationships and physical activity behavioral intentions.

### 2.4. Chain mediation of social support and exercise of self-efficacy

According to the above research review, it was found that college students' peer relationships, social support, and exercise self-efficacy have an important influence on the development of physical activity behavioral intention, in which social support and exercise self-efficacy play a mediating role in the process of peer relationships' influence on exercise self-efficacy. Meanwhile, some studies have shown that there is a correlation between social support and self-efficacy [37] and that social support can influence self-efficacy, and when college students feel support and encouragement from peers, family members, teachers, and other groups, they can complete the challenges of school learning and life with more confidence, which enhances their self-efficacy [38]. Other studies have also found that in the school setting, college students can gain social support through positive peer relationships, which enhances self-efficacy and gives them more confidence to participate in activities and complete tasks [39]. In school physical activity, individuals gain some support during exercise by working with close peers, further increasing their exercise self-efficacy, and generating positive physical activity behavioral intentions, leading to more active participation in physical activity. Based on this, the present study proposes Hypothesis H4: Social support and exercise self-efficacy may play a chain-mediated role in the influence of peer relationships on physical activity behavioral intentions (Fig 1).

## 3. Materials and methods

### 3.1. Participants

In this study, before the questionnaire was administered, the minimum sample size was calculated by using the G*Power program, setting α=0.05,β=0.80 and Effect Size=0.15 to obtain the minimum sample size required for the study, which was 77.This study through the convenience sampling of a province of four colleges and universities as the survey object, to ensure the right to information and follow the principle of voluntariness of the subjects, the implementation of the test before informing the subjects to fill in the content of the confidentiality programmer, after agreeing to the questionnaire through the questionnaire star platform from June 3 to June 7, 2024, issued time questionnaires 589(03/06/2024 to 07/06/2024), recovered 514 valid questionnaires, the effective recovery rate of 87.27%, the age and standard deviation was M ± SD = 19.00 ± 1.27, of which 285 male students, 229, accounting for 44.4% (Table 1).

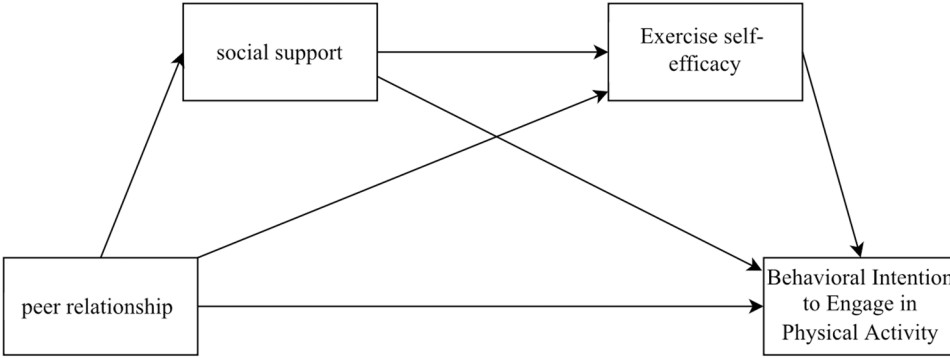

**Fig 1. Conceptual architecture model.**

**Table 1. Distribution of demographic variables among survey respondents.**

| Variable | Form | Numbers | Precent (%) |
|---|---|---|---|
| Gender | Male | 285 | 55.4 |
| | Female | 229 | 44.6 |
| Grade | First year of university | 173 | 33.7 |
| | Second year of university | 170 | 33.1 |
| | Third year of university | 171 | 33.3 |
| Age | 17 years old | 75 | 14.6 |
| | 18 years old | 113 | 22.0 |
| | 19 years old | 138 | 26.8 |
| | 20 years old | 112 | 21.8 |
| | 21 years and over | 76 | 14.8 |
| add up the total | | 514 | 100 |

**3.1.1. Ethics statement.** All methods were carried out according to relevant guidelines and regulations. The studies involving human participants were reviewed and approved by the Chengdu Sport University ethics committee (Ethical Approval Number: 202494). The participants provided their written informed consent to participate in this study.

## 3.2. Methods

**3.2.1. Peer relationship scale.** The Peer Relationship Scale prepared by Wei Yunhua was used [40], which contains 20 questions, the scale is based on a 5-point Likert scale, and each entry has 5 levels ((1 = not at all correct; 2 = incorrect; 3 = uncertain; 4 = correct; 5 = completely correct), and the higher the score is, the higher the subject's peer relationship is, and the reliability of this scale is good. In the present study, the Cronbach α coefficient was 0.857.

**3.2.2. Physical activity behavioral intention scale.** The scale of college students' physical activity behavior was selected from the physical activity intention subscale of the Physical Activity Rating Scale compiled by Liang Deqing et al. [41], with a total of 12 questions and a 5-point Likert scale, with 5 levels for each entry (1 = very non-compliant; 2 = non-compliant; 3 = unsure; 4 = compliant; and 5 = very much compliant), and the higher the score represents the higher the intention to be physically active, and the sub scalar scale questionnaire had good reliability and validity, and the Cronbach α coefficient in this study was 0.808.

**3.2.3. Social support scale.** The Social Support Rating Scale developed by Ye Yuemei and Dai Xiaoyang was used [42], is based on Shao Shuiyuan's theoretical model of social support, and includes three factors, subjective support, objective support, and support utilization, and consists of total of 17 questions, with a 5-point Likert scale (1 = not at all, 2 = not at all, 3 = unsure, 4 = in line with, and 5 = in full line with). Higher scores on the scale indicate higher levels of social support. In the current study, the Cronbach α coefficient was 0.880.

**3.2.4. Exercise self-efficacy scale.** The Chinese version of the Exercise Self-Efficacy Scale [43], developed by Motl et al. and revised by our scholars Chen et al. The scale consists of 8 questions and is scored on a 5-point Likert scale, ranging from '1' for complete disagreement to '5' for complete agreement. The higher the sum of the scores for all questions, the higher the students' self-efficacy for exercise. In this study, the Cronbach's alpha coefficient was 0.762.

**3.2.5. Data analysis.** The data collected in this study were processed and analyzed using SPSS 27.0, common method bias test, descriptive statistics, correlation analysis, independent samples t-test, and analysis of variance (ANOVA) through SPSS; the process macro program developed by Hayes was used, and Model 6 was selected, where M1 = Social support, M2 = Exercise self-efficacy. X = Peer Relationships, Y = Physical Activity Behavioral Intention, Bootstrap Samples = 5000.

## 4. Results

### 4.1. Common methodology bias test

The current study used the Harman one-way test for common method bias [44]. Unrotated principal component analysis of peer relationships, social support, exercise self-efficacy, and physical activity behavioral intention was conducted using SPSS27.0, resulting in the extraction of a total of 14 factors with an eigenroot greater than 1. Among them, the first factor cumulatively explained 27.04% of the total variance, which was less than the 40% criterion, indicating that there was no serious common method bias in this study.

### 4.2. A current analysis of peer relationships, social support, exercise self-efficacy and physical activity behavioral intentions of college students

A one-sample t-test was conducted on the sample data, set to 3 as the median, and the results showed (Table 2) that the scores of college students' peer relationships, social support, exercise self-efficacy, and physical activity behavioral intentions were only slightly higher than the median of 3. This indicates that, although college students' peer relationships peer relationships, social support, exercise self-efficacy, and physical activity behaviors did not show any significant deficiencies, the degree of development was average and still needs to be further improved.

### 4.3. Correlation analysis of variables

Pearson's correlation analysis of peer relationship, social support, exercise self-efficacy and physical activity behavioral intention was conducted using SPSS27.0. The results of descriptive statistics showed (Table 3) that peer relationships were positively correlated with social support, exercise self-efficacy, and physical activity behavioral intention ($p < 0.01$), and social support was positively correlated with exercise self-efficacy and physical activity behavioral intention ($p < 0.01$), and exercise self-efficacy was positively correlated with physical activity behavioral intention ($p < 0.01$).

**Table 2. One-sample t-tests for peer relationships, social support, exercise self-efficacy, and physical activity behavioral intentions.**

|  | M | SD | T |
| --- | --- | --- | --- |
| 1. Peer relationship | 3.04 | 0.52 | 1.68 |
| 2. Social support | 3.07 | 0.58 | 2.73 |
| 3. Exercise self-efficacy | 3.09 | 0.57 | 3.40 |
| 4. Behavioral Intention to Engage in Physical Activity | 3.10 | 0.63 | 3.63 |

**Table 3. Correlation coefficient between variables.**

|  | 1 | 2 | 3 | 4 |
| --- | --- | --- | --- | --- |
| 1. Peer relationship | 1 |  |  |  |
| 2. Social support | .640** | 1 |  |  |
| 3. Exercise self-efficacy | .832** | .651** | 1 |  |
| 4. Behavioral Intention to Engage in Physical Activity | .743** | .578** | .758** | 1 |

Note : '
*' indicates p＜0.05, '
**' indicates p＜0.01.

## 4.4. Analysis of differences in variables by gender and academic level

According to the central limit theorem, the distribution of sample means can be considered to converge to a normal distribution when the sample size is greater than 200 [45,46]. The sample size of this study amounted to 514 and the skewness (0.002/0.007) and kurtosis (-1.023/-1.509) of the age and learning strata were significantly lower than the critical values (skewness < 3 and kurtosis < 10) proposed by West et al. (1995), which is in line with the assumption of normality for parametric tests [47].

In this study, independent samples t-tests were conducted on the gender of the subjects (as shown in Table 4), and Levene's variance equivalence test for gender showed that there were no significant differences (P > 0.05) in social support, exercise self-efficacy, and physical activity behavioral intention, so variance chi-squared data were used, and there were significant differences in peer relationships, and variance non-chi-squared data were used. In the mean equivalence t-test, there is no significant difference in peer relationships and social support (P > 0.05), indicating that there is no gender difference in peer relationships and social support among college students; however, there is a significant difference in exercise self-efficacy and intention to engage in physical activity behaviors (P < 0.001), indicating that there is a gender difference in exercise self-efficacy and intention to engage in physical activity behaviors among college students, and through further comparison It was found (as in Table 5) that the scores of male students were higher than those of female students.

After further comparison (as shown in Table 5), the scores of male students were higher than those of female students. Meanwhile, when ANOVA was conducted on academic segments and ages (as shown in Table 6), there were no significant differences in academic segments and ages (P > 0.05) for peer relationships, social support, exercise self-efficacy, and intention to engage in physical activity behaviors, suggesting that there are no differences in age and academic segments for peer relationships, social support, exercise self-efficacy and intention to engage in physical activity behaviors among college students.

## 4.5. Mediation effect test

College students' peer relationship, social support, exercise self-efficacy, and physical activity behavior intention are significantly correlated with each other, which is in line with the statistical requirements of direct and indirect effect analysis

**Table 4. Independent samples t-test for gender.**

| Variant | Implicit variable | HV-test | Levine's test of variance equivalence | | Mean equivalence t-test | |
|---|---|---|---|---|---|---|
| | | | F | P | t | P |
| Gender | 1. Peer relationship | Variance non-chirality | 4.434 | 0.036 | 1.629 | 0.104 |
| | 2. Social support | Variance chirality | 0.053 | 0.819 | 1.309 | 0.191 |
| | 3. Exercise self-efficacy | Variance chirality | 1.307 | 0.254 | 3.998 | <0.001 |
| | 4. Behavioral Intention to Engage in Physical Activity | Variance chirality | 0.433 | 0.511 | 5.166 | <0.001 |

**Table 5. Analysis of high and low gender difference scores.**

| Variant | Gender | Number | Mean (M) | Standard deviation (SD) |
|---|---|---|---|---|
| Exercise self-efficacy | Male | 285 | 3.175 | 0.545 |
| | Female | 229 | 2.975 | 0.587 |
| Behavioral Intention to Engage in Physical Activity | Male | 285 | 3.225 | 0.617 |
| | Female | 229 | 2.945 | 0.605 |

of peer relationship and physical activity behavior intention, while peer relationship is the independent variable, physical activity behavior intention is the dependent variable, social support, and exercise self-efficacy are the mediator variables, based on controlling demographic variables. Hayes' process macro program was used to test and analyze the mediating role of social support and exercise self-efficacy between peer relationships on physical activity behavioral intentions among university students according to Bootstrap, and the values of the paths between the variables are shown in Fig 2. Before the regression analysis, the test of multicollinearity was conducted, and the results of the statistical analysis of covariance, using physical activity behavioral intention as the dependent variable, showed that the VIF of peer relationship, social support and exercise self-efficacy were 3.381, 1.814 and 3.581, respectively, which were less than 5, and there was no serious multicollinearity, and the results were acceptable. In addition, an interaction analysis was conducted and found that no significant effects were found in the interaction analysis between social support, peer relationships, and exercise self-efficacy; therefore, the main effects model was supported. The steps of the regression model (Table 7) were (1) using intention to engage in physical activity behavior as the dependent variable, and gender, school year, and age as the predictor variables, controlling for their possible effects on college students' physical activity behavior, and inputting peer relationships to establish a regression model to examine the total effect of peer relationships on intention to engage in physical activity behavior after controlling for the variables, and it was found that college students' peer relationships were able to positively predict the intention to engage in physical activity behavior ($\beta = 0.730$, $p < 0.001$). (2) Social support and exercise self-efficacy were added sequentially to the model to test whether they mediated the influence of peer

**Table 6. ANOVA for grade and age.**

| Grouping variable | Implicit variable | Mean square | F | P |
|---|---|---|---|---|
| Grade | 1. Peer relationship | 0.266 | 0.990 | 0.372 |
| | 2. Social support | 0.520 | 1.543 | 0.215 |
| | 3. Exercise self-efficacy | 0.329 | 1.004 | 0.367 |
| | 4. Behavioral Intention to Engage in Physical Activity | 0.006 | 0.015 | 0.985 |
| Age | 1. Peer relationship | 0.219 | 0.815 | 0.516 |
| | 2. Social support | 0.345 | 1.020 | 0.396 |
| | 3. Exercise self-efficacy | 0.387 | 1.182 | 0.318 |
| | 4. Behavioral Intention to Engage in Physical Activity | 0.150 | 0.380 | 0.823 |

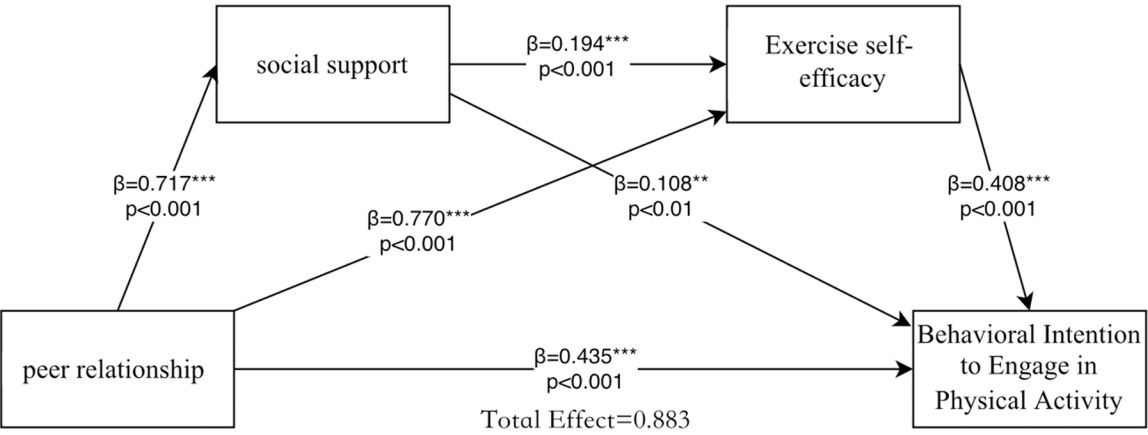

**Fig 2. Pathways of peer relationship influence on behavioral intentions to engage in physical activity.**

relationships on intention to engage in physical activity and whether a chain mediation effect could be formed, and it was found that peer relationships positively predicted intention to engage in physical activity (β=0.359, p<0.001), and social support positively predicted intention to engage in physical activity (β=0.730, p<0.001). exercise intention (β = 0.101, p < 0.001), and exercise self-efficacy significantly predicted physical activity behavior intention (β = 0.372, p < 0.001). The above data suggest that college students' peer relationships may indirectly predict physical activity behavior intention through social support and exercise self-efficacy.

The mediating effects of social support and exercise self-efficacy in college students' peer relationships and physical activity behavioral intentions were analyzed using the bootstrap method (e.g., Table 8), with a sample size of 5000 selected and 95% confidence intervals used, which showed that peer relationships directly predicted physical activity behavioral intentions and that social support and exercise self-efficacy mediated the mediating effects (with a total of three paths), with a total mediation effect of 0.448, and a bootstrap 95% confidence interval for the mediation effect of [0.302,0.561], indicating a significant mediation effect (not exceeding 0). The first path: peer relationship → social support → physical activity behavioral intention, the effect value is 0.078, accounting for 8.83% of the total effect; the second path: peer relationship → exercise self-efficacy → physical activity behavioral intention, the effect value is 0.314, accounting for

Table 7. Regression analysis of peer relationships, social support, and exercise self-efficacy on physical activity behavioral intentions.

| variant | Behavioral Intention to Engage in Physical Activity (1) | | Behavioral Intention to Engage in Physical Activity (2) | | Behavioral Intention to Engage in Physical Activity (3) | |
|---|---|---|---|---|---|---|
| | β | t | β | t | β | t |
| Gender | -0.171 | -5.934*** | -0.17 | -6.015*** | -0.128 | -4.674*** |
| Grade | -0.042 | -0.984 | -0.054 | -1.293 | -0.059 | -1.493 |
| Age | 0.032 | 0.758 | 0.033 | 0.787 | 0.032 | 0.797 |
| peer relationship | 0.73 | 25.374*** | 0.619 | 16.885*** | 0.359 | 7.185*** |
| social support | | | 0.174 | 4.742*** | 0.101 | 2.763** |
| Exercise self-efficacy | | | | | 0.372 | 7.247*** |
| $R^2$ | 0.582 | | 0.6 | | 0.637 | |
| $\Delta R^2$ | 0.529 | | 0.018 | | 0.038 | |
| F | 177.21*** | | 152.25*** | | 148.49*** | |

Note : '

'**' indicates p<0.01, '

'***' indicates p<0.001.

Table 8. Explanatory table for total, direct and indirect effects.

| Effect | Path | Efficacy value | SE | LLCI | ULCI | Effective quantity |
|---|---|---|---|---|---|---|
| Total effect | | 0.883 | 0.035 | 0.815 | 0.952 | 100% |
| Direct effect | Direct path | 0.435 | 0.061 | 0.316 | 0.554 | 49.26% |
| Total indirect effect | | 0.448 | 0.067 | 0.302 | 0.561 | 50.74% |
| Indirect effect | Path1 | 0.078 | 0.034 | 0.019 | 0.151 | 8.83% |
| | Path2 | 0.314 | 0.055 | 0.195 | 0.413 | 35.56% |
| | Path3 | 0.057 | 0.019 | 0.025 | 0.098 | 6.46% |

Note: Path 1: peer relationships → social support → physical activity behavioral intentions; Path 2: peer relationships → exercise self-efficacy → physical activity behavioral intentions; Path 3: peer relationships → social support → exercise self-efficacy → physical activity behavioral intentions.

35.56% of the total effect; the third path peer relationship → social support → exercise self-efficacy → physical activity behavioral intention, the effect value is 0.057, accounting for 35.56% of the total effect. value was 0.057, accounting for 6.46% of the total effect.

## 5. Discussion

### 5.1. Discussion of gender differences in exercise self-efficacy and physical activity behavioral intention

Compared with other studies, the present study further found, based on previous research, that there were gender differences in exercise self-efficacy and physical activity behavioral intention in the college student population, and that male students scored higher than female students. This difference may stem from the interaction between socio-cultural constructs and individual practical experience. First, gender role theory suggests that gender role expectations (e.g., "being strong" is more often seen as a core element of masculinity) may be influenced through differential socialization processes. That is, males receive early access to athletic resources, whereas females may inhibit the expression of behavioral intentions due to "stereotype threat" [48]. Second, the gradient of accumulated exercise experience exacerbates this difference; in this paper, men had higher exercise self-efficacy than women, and according to Bandura's theory of self-efficacy, the sustained successful experience reinforces the virtuous circle of individual exercise confidence, and thus, men had higher exercise self-efficacy and physical activity behavioral intention than women [49]. Finally, structural environmental factors should not be overlooked. Implicit gender segregation of campus sports spaces (e.g., centralized layout of basketball and soccer courts) may weaken women's sense of belonging to mainstream sports, and thus they face higher psychological costs of participating in physical activity, and their sense of athletic self-efficacy and behavioral intentions to engage in physical activity may be affected. This finding suggests that colleges and universities need to break through the single explanatory framework of physiological differences in their future sports promotion strategies and focus on deconstructing gender barriers in sociocultural constructs, such as developing gender-inclusive physical education and sports curricula and establishing a system of good female role models for sport, to improve the gender differences in self-efficacy for sport and intention to engage in physical activity behaviors.

### 5.2. Significant direct effects of peer relationships on physical activity behavioral intentions among university students

The results of the current study showed that college students' peer relationships significantly and positively predicted physical activity behavioral intention, with an effect size of 0.435, and Hypothesis H1 was confirmed. Building on previous findings that peer relationships influence physical activity, peer relationships were found to influence physical activity behavioral intentions [14,50]. First, from the perspectives of social cognitive theory and environmental perception theory, college students' perception and understanding of the campus environment (e.g., peer relationships) are internalized into discriminative information about the behavioral environment, which stimulates emotions and establishes good physical activity behavioral intentions [51]. Second, research has shown that positive peer relationships can be a source of emotional resonance and support acquisition for college students, which can help them enhance their self-beliefs and guide the practice of related activities in their life and learning interactions [52]. In addition, according to the Hierarchy of Needs Theory, when college students feel positive feedback (e.g., acceptance and encouragement, etc.) from their peers in the physical activity context in the campus environment, they will show a more positive state and gain a pleasurable experience in the process, which further enhances their physical activity behavioral intention [53]. In the field of school education and the context of quality education, physical education teaching activities can not only enhance the relationship between college students but also, good peer relationships can help them to be more positive in their physical exercise behavioral intentions. As previous researchers have found in the process of studying peer effects, the performance and development of individual behaviors are influenced by their peers, and at the same time, they are also generated by their interaction

with the external environment [54]. Therefore, to help college students establish correct sports values, colleges and universities need to enhance individuals' physical activity behavioral intentions and promote their overall physical and mental development by fostering positive peer relationships and creating supportive environments.

### 5.3. The mediating role of social support between peer relationships and physical activity behavioral intentions

The results of the mediation effect showed that social support plays a partial mediating role in peer relationships and physical activity behavioral intention (relative mediation effect of 8.83%), which confirms hypothesis H2 of this study, i.e., peer relationships not only can influence physical activity behavioral intention but also can further act on physical activity behavioral intention through social support. Peer relationships are an important part of the social network structure of college students. On the one hand, good peer relationships can promote the acquisition of social support for college students [26], when individuals feel care and acceptance in the process of interacting with peers, they are more willing to communicate with peers and can obtain support and help from peers. Meanwhile, attachment theory suggests that when individuals have good peer relationships, they are more likely to receive support from their peers in the process of seeking help from them, thus helping them to cope with difficulties and pressures from the outside world [55]. On the other hand, college students' physical activity behaviors are influenced by social support, exercise atmosphere, and self-efficacy [56]. In the developmental process of college students, they face the problem of adapting from high school to college, and due to the changes in their surroundings, the support they receive in school comes more from their classmates and friends, whose support can provide direction for the individual and promote college students' participation in physical activity [23]. Meanwhile, from the perspective of behavioral change theory, college students can change their behavioral intentions and participate more actively in physical activity by using the help provided by the people around them after receiving social support [57]. Previous research has also demonstrated this idea that social support can influence physical activity behavior [58]. This study also found that the intention to engage in physical activity behavior, as an antecedent of physical activity behavior, was also influenced by social support, further enriching existing research.

### 5.4. The mediating role of exercise self-efficacy between peer relationships and physical activity behavioral intentions

The study showed that peer relationships not only directly affect physical activity behavioral intention, but also further predict physical activity behavioral intention through exercise self-efficacy, which also proved hypothesis H3 of this study, and further enriched the correlation between peer relationships, exercise self-efficacy and physical activity behavioral intention based on the findings of the previous studies. Cognitive theory suggests that college students' experiences and evaluations in peer relationships are one of the important sources of their self-efficacy acquisition [59]. It has been shown that students entering college are more inclined to seek and obtain emotional understanding and support from their peers to gain more psychological resources to enhance their self-efficacy [60]. In the schooling environment, individuals can obtain encouragement and assistance from their peers through good peer relationships and enhance exercise self-efficacy to be more actively involved in physical activity [61]. Meanwhile, the ternary interaction theory suggests that individual's behavior is influenced by personal, behavioral and environmental influences, and that college students with higher exercise self-efficacy are able to generate stronger intentions to engage in physical activity behaviors in their school life, which in turn provides clearer motivation to participate in physical activity and generate exercise behaviors [62]. Related studies have also further indicated that exercise self-efficacy can improve college students' enthusiasm for physical activity and increase the persistence of their physical activity behavior [63]. Thus, college students with positive peer relationships are able to obtain higher exercise self-efficacy during physical exercise and generate stronger intentions for physical exercise behaviors.

 

## 5.5. Chain mediation of social support and exercise self-efficacy

In this study, in the process of examining the influence of college students' peer relationship, social support, and exercise self-efficacy on the intention of physical exercise behavior, it was found that the total effect was 0.883, in which there was a two-by-two positive correlation between peer relationship, social support, exercise self-efficacy, and physical exercise behavior, with the direct effect of β=0.435, and the total indirect effect of β=0.448, which can be seen that the mediator effect is in the model The third mediation path (peer relationship→social support→exercise self-efficacy→intention of physical activity behavior) accounted for 6.46% of the total effect, indicating that the chain mediation effect was established, and the peer relationship could not only influence college students' intention of physical activity behavior through the simple mediation of social support and exercise self-efficacy, but also influence college students' intention of physical activity behavior through the chain mediation effect to influence physical activity behavioral intention. First, self-efficacy suggests that in social environments, individuals increase their self-efficacy through their own experiences, alternative experiences others' evaluations, and self-states [64], which is similar to the results of this study. In the university campus environment, college students improve their exercise self-efficacy through external support from teachers, classmates, and other sources, which helps them overcome difficulties and challenges in exercise. Second, college students with high social support tend to integrate better into their class and dormitory groups, which leads to more positive feedback and increased exercise self-efficacy in their daily life and exercise [38]. Based on previous studies, this study further analyzed the intrinsic relationship of peer relationships on physical activity behavioral intention, proposed and proved the chain mediating role of social support and exercise self-efficacy between the two, which is conducive to college students' recognition of the internal influence mechanism of peer relationships on physical activity behavioral intention, and is of guiding significance to the development of school sports and students' physical and mental health development.

Finally, this study found no significant difference in age between peer relationships, social support, exercise self-efficacy, and physical activity behavioral intention, probably because the study group was mainly college students, whose age gap was smaller, and individuals in this age group were closer in physiological and psychological development. The social-ecological system theory emphasizes that individual development is influenced by multi-level environmental factors, including micro-systems (e.g., family, school), meso-systems (e.g., community), and macro-systems (e.g., culture, policy) [65]. Based on this theory, the behavior and psychological development of individuals are not only influenced by their factors, but also by the interaction of the surrounding environment. For college students, they are influenced by similar (schooling, socio-cultural, etc.) environments in the same socio-ecological system, and these similar environmental factors may mask the effects of age on peer relationships, social support, exercise self-efficacy, and physical activity behavioral intentions, resulting in a non-significant age effect.

## 6. Conclusion

This study demonstrated a strong relationship between college students' peer relationships, social support, exercise self-efficacy, and physical activity behavioral intentions. First, college students' peer relationships significantly and positively predicted intentions to engage in physical activity behavior, which is an important factor in promoting college students' physical and mental health. Second, social support and exercise self-efficacy not only play a simple mediating role in the influence of peer relationships on physical activity behavioral intention, but also influence physical activity behavioral intention through the chain effect of social support and exercise self-efficacy.

Therefore, based on the findings of the study, this paper suggests that in order to strengthen college students' peer relationships and promote physical activity and physical activity behavioral intentions, and then promote their overall physical and mental development, colleges and universities can take the following measures: first, organize team sports activities, such as basketball, soccer and other competitions, to enhance teamwork and cohesion; second, set up sports clubs and interest groups to provide a platform for students to participate in sports together Thirdly, to carry out sports training

and lectures to improve students' motor skills and sense of self-efficacy; fourthly, to provide adequate sports facilities and resources to ensure that students have good conditions for exercising; fifthly, to strengthen mental health education to help students establish a correct outlook on life and values; and sixthly, to promote peer support, encouraging students to help and encourage each other through trainings and activities.

Although this study revealed a close relationship between peer relationships, social support, exercise self-efficacy, and physical activity behavioral intentions among college students, there are some shortcomings, and future research can be improved in the following areas:

1. The fact that this study was cross-sectional and used undergraduate students as the survey population may limit inferences about the causal relationships among the variables. Future research could adopt a longitudinal research design, expand the sample size, and examine the intrinsic mechanisms between the variables in both the longitudinal and cross-sectional dimensions, e.g., undergraduates in different grades could be tracked over time to better understand the trends of these variables over time. In particular, the method of expanding the sample size: although the sample size of this study is 514, which is far above the minimum sample size, in order to further improve the generalizability and reliability of the study, future studies can further expand the sample size to include different regions and types of colleges and universities, for example, stratified sampling by "stage of education", to compare the differences in mediating effects among different developmental stages, and to examine the mediating effects of different developmental stages. Compare the differences in mediation effects at different stages of development.

2. While social support and exercise self-efficacy undoubtedly play a pivotal mediating role in examining the relationship between peer relationships and college students' physical activity behavioral intentions, there may be other potential mediating variables that also exert influence. These include an individual's self-perception, subjective exercise experience, and class belonging. In addition, it is essential to control for confounding variables such as socioeconomic status, exercise experience, and personality traits. Subsequent research can be based on this study, starting from the policy context, focusing on the physical and mental health development of adolescents, combining the disciplinary strengths and characteristics of school sports, and considering the influence of internal and external factors on adolescents' physical exercise behavioral intentions. In this way, programs and paths can be provided for promoting the comprehensive physical and mental development of adolescents. For instance, it would be feasible to examine the interplay among familial environment, institutional policy, and societal culture in influencing the physical activity patterns of college students.

3. Based on the results of the study, future interventions, such as strengthening mental health education, optimizing the physical education curriculum, and providing more social support, can be designed, and implemented to promote the physical and mental health development of college students. An intervention study can be conducted to observe the effects of measures such as providing psychological counseling and increasing opportunities for physical activity on college students' behavioral intentions to engage in physical activity.

## Supporting information

**S1 Data. Data availability statement.**
(XLSX)

## Author contributions

**Conceptualization:** Xielin Zhou.

**Data curation:** Xielin Zhou.

**Formal analysis:** Xielin Zhou.

**Methodology:** Xielin Zhou.

Project administration: Xielin Zhou.

Software: Xielin Zhou, Mu Zhang.

Writing – original draft: Xielin Zhou.

Writing – review & editing: Xielin Zhou, Mu Zhang, Lu Chen, Bo Li, Jiao Xu.

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
