## [Decision Letter · Decision Letter 0]

31 Jan 2025

PONE-D-24-44384The effect of peer relationships on college students' behavioral intentions to be physically active: the chain-mediated role of social support and exercise self-efficacyPLOS ONE

Dear Dr. zhang,

Thank you for submitting your manuscript to PLOS ONE. After careful consideration, we feel that it has merit but does not fully meet PLOS ONE’s publication criteria as it currently stands. Therefore, we invite you to submit a revised version of the manuscript that addresses the points raised during the review process.

We look forward to receiving your revised manuscript.

Kind regards,

Jianpeng Fan

Academic Editor

PLOS ONE

Additional Editor Comments:

In this manuscript, the authors have made corresponding efforts. The reviewers’ evaluations of the manuscript are not uniform, but I believe the authors should be given another opportunity to revise and improve it. The authors need to carefully address the reviewers’ comments, especially by addressing the theoretical gaps in existing research to elucidate the research value and significance of this study.

Reviewers' comments:

Reviewer's Responses to Questions

**Comments to the Author**

1. Is the manuscript technically sound, and do the data support the conclusions?

Reviewer #1: Yes

Reviewer #2: Yes

Reviewer #3: Yes

2. Has the statistical analysis been performed appropriately and rigorously? 

Reviewer #1: Yes

Reviewer #2: Yes

Reviewer #3: No

3. Have the authors made all data underlying the findings in their manuscript fully available?

Reviewer #1: Yes

Reviewer #2: Yes

Reviewer #3: No

4. Is the manuscript presented in an intelligible fashion and written in standard English?

Reviewer #1: Yes

Reviewer #2: Yes

Reviewer #3: No

5. Review Comments to the Author

Reviewer #1: This is a well articulated and well researched study. Its use of large sample size gives more credibility to the findings, as this can help reduce the subjectiveness of self-report.

Your exploration of gender differences is also very commendable. However, I think you can delve more into why men outperformed women in exercise self-efficacy and intentions to engage in physical activity. For instance, could this be due to gender roles or societal bias (or other factors) whereby some societies expect men to be more physically fit than women and regard women as weaklings.

Furthermore, you may need to define the term “good peer relationship” to further contextualize your work. For instance, on number 75 and following numbers, you wrote, 75 when studying issues related to physical activity behaviors [7, 8]. Most studies have found that good 76 peer relationships (?????????????????). It might help to define this concept even if it means putting a synonym in a bracket following it. It is important because this term may be relative. And what is good for a student may be bad for another.

Fair enough that you reported the age of your participants. However, you can further enhance the study by reporting the effects of age, just as you did for gender. This is important because age can be a determining factor in peer relationship and behavioural intention. For instance, some persons may feel reluctant to associate with people outside their age bracket. So does age play any role in peer relationship in your study? Also age may be an influencing factor in students behavioural intentions to engage in physical activities. For instance, priorities may change due to age (older students may begin to prioritize academic/work obligations compared to the much younger ones).

Reviewer #2: The study is well-designed and its objectives are clear; however, it has certain shortcomings that can be addressed in the review process. These include improving sample representation, expanding the theoretical framework, enhancing statistical analysis, and providing more specific and actionable recommendations.

1) The sample consists of only 514 students. While this number may be sufficient for certain statistical analyses, it does not adequately capture the diversity of university students, particularly across different cultures or universities with varied educational programs.

2) The study is based on theories; however, it does not provide sufficient details on how these theories practically support the hypotheses.

3) a- The study did not clarify whether the normality of the data distribution was verified using tests such as Shapiro-Wilk or Kolmogorov-Smirnov, which is essential when employing tests like t test or analysis of variance (ANOVA).

b- Regression analysis was used to determine the relationship between variables; however, it was not clarified whether multicollinearity among the independent variables was assessed.

c- Interaction analysis between independent variables (e.g., social support and peer relationships) was not conducted to determine whether the effects depend on the levels of other variables.

4) Although the limitations were mentioned, no clear solutions or future research plans were proposed to address them.

Reviewer #3: Abstract

1. The following sentences are lengthy and complex, making comprehension difficult.

"The behavioral intentions to physical activity have a positive impact on an individual's physical and mental health, social development, social adaptation, and study and work, and are also important in an individual's overall development."

2. In line 25, what does the author mean by the term ''support''? Does it mean social support?

introduction

3. The statement of the problem failed to justify why this study was conducted. The research problem needs major revisions; it has to address the novelty of this study and why this study was conducted.

4. The introduction is too long. Why should H1 be examined in the current study if it has already been discussed in previous studies?

5. However, similar studies exist exploring peer influence on physical activity. The novelty must be clearly articulated—what does this study add beyond existing work?

6. No control for confounders—e.g., socioeconomic status, prior sports experience, or personality traits, which may also influence exercise self-efficacy.

Results

7. Ambiguities in Results Presentation:

It is not clear from which scale or questionnaire mobile phone addiction was derived.

In Table 3 (Correlation Coefficients), the sentence "mobile phone addiction was negatively correlated with physical activity behavior" seems out of context.

Figure 2 refers to "Pathways of mobile phone addiction on physical activity behavior."

Discussion

8. The discussion and conclusion are very poorly written. No comparisons are made between this study and other studies.

9. Discussion & Practical Implications (Weak to Moderate)

10. How can universities practically strengthen peer relationships to improve physical activity?

11. Are there specific intervention programs that could be developed based on the findings?

6. PLOS authors have the option to publish the peer review history of their article (what does this mean? ). If published, this will include your full peer review and any attached files.

**Do you want your identity to be public for this peer review?** For information about this choice, including consent withdrawal, please see our Privacy Policy .

Reviewer #1: No

Reviewer #2: **Yes: ** Prof. Mohammad Alshumrani

Reviewer #3: No

---

## [Author Response · Author response to Decision Letter 1]

6 Feb 2025

Responses to Editor and Reviewers

Dear Editor and all the Reviewers,

Happy Chinese New Year. Thank you for your comments concerning our manuscript entitled “The Effect of Peer Relationships on College Students' Behavioral Intentions to Be Physically Active: the chain-mediated Role of Social Support and Exercise Self-efficacy”(ID: PONE-D-24-44384).

Thank you for your professional review work, constructive comments, and suggestions on our manuscripts. These comments helped to improve the academic rigor of our articles, and they were beneficial for revising and improving our paper and for highlighting the significance of our research. We have studied the comments carefully and have made corrections that we hope meet with approval.

Based on the comments of the 3 reviewers, we have made corresponding changes, which are marked as highlighted blue in the manuscript for Reviewer 1, light blue for Reviewer 2, and highlighted in yellow for Reviewer 3.

Your sincerely Zhouxielin; ZhangMu; Chen Lu; Li Bo; Xu Jiao

2025/2/5

Editor#

Additional Editor Comments:

In this manuscript, the authors have made corresponding efforts. The reviewers’ evaluations of the manuscript are not uniform, but I believe the authors should be given another opportunity to revise and improve it. The authors need to carefully address the reviewers’ comments, especially by addressing the theoretical gaps in existing research to elucidate the research value and significance of this study.

Responses to Editor:

We are very grateful for the comments given by the editorial staff, which allowed us to understand the shortcomings of the article and to be able to correct them.

Theoretical support was added to this study. In the introduction, Bandura's Ternary Interaction Theory was used to support why peer relationships, social support, exercise self-efficacy, and physical activity behavioral intention were examined. Bandura’s Ternary Interaction Theory suggests that environmental factors (e.g., peer relationships, social support) form a cyclical dynamic with behavioral intention through personal cognition (e.g., self-efficacy). Based on the ternary interaction theory, this study introduces, peer relations and social support among environmental factors and exercise self-efficacy among personal cognitive factors to explore the influence mechanism of physical activity behavioral intention, aiming at enriching the research results in the existing field, and providing theoretical support for the development of physical education and lifelong physical education in colleges and universities.

In addition, in the first part of the research hypothesis, we added “The theory of planned behavior suggests that an individual's behavior is influenced by his or her behavioral intentions, which in turn are influenced by behavioral attitudes, subjective norms, and perceived behavioral control. Through this theory, peer relationships may influence individuals' physical activity behavioral intentions by influencing their behavioral attitudes and perceptual behavioral control.” To support the proposal of H1.

In the second part of the research hypothesis, we added further attachment theory to support the influence of peer relationships on social support. “On the other hand, John Bowlby's attachment theory emphasizes that individuals seek closeness and attention when forming attachment relationships with others to gain a sense of security and self-confidence, which continues to influence their social behavior and emotion regulation, and from the perspective of attachment theory, the attachment relationships that college students form with their peers during their participation in school-related activities can have an emotions and behaviors in an important way.”

In the third part of the hypothesis, we added Albert Bandura's self-efficacy theory to support the influence of exercise self-efficacy on physical activity behavioral intentions. “Self-efficacy theory (Albert Bandura) suggests that self-efficacy has a significant impact on an individual's behavioral choices, level of motivation, emotional state, and adherence to behavior. Specifically, when individuals have higher self-efficacy, they are more likely to choose challenging tasks and show higher levels of persistence and effort in the face of difficulties.”

In addition, we have further improved the content of the data analysis, the conclusions and recommendations, and the outlook of the study based on the comments of the reviewers.

Finally, I would like to thank again the editorial staff and the reviewers for their suggestions, which enabled us to learn more and improve our research. Happy Chinese New Year to you all.

Reviewer#1:

Responses to comment: (Revisions are highlighted in blue, for example AAA)

Q1: However, I think you can delve more into why men outperformed women in exercise self-efficacy and intentions to engage in physical activity. For instance, could this be due to gender roles or societal bias (or other factors) whereby some societies expect men to be more physically fit than women and regard women as weaklings.

A1: We are very grateful to the reviewers for this comment, which helps us to further refine the conclusions of the study and make it more valuable. Based on this issue, we added an analysis of gender. The modifications are as follows

Discussion of Gender Differences in Exercise Self-Efficacy and Physical Activity Behavioral Intention

Compared with other studies, the present study further found, based on previous research, that there were gender differences in exercise self-efficacy and physical activity behavioral intention in the college student population, and that male students scored higher than female students. This difference may stem from the interaction between socio-cultural constructs and individual practical experience. First, gender role theory suggests that gender role expectations (e.g., “being strong” is more often seen as a core element of masculinity) may be influenced through differential socialization processes. That is, males receive early access to athletic resources, whereas females may inhibit the expression of behavioral intentions due to “stereotype threat”. Second, the gradient of accumulated exercise experience exacerbates this difference; in this paper, men had higher exercise self-efficacy than women, and according to Bandura's theory of self-efficacy, the sustained successful experience reinforces the virtuous circle of individual exercise confidence, and thus, men had higher exercise self-efficacy and physical activity behavioral intention than women. Finally, structural environmental factors should not be overlooked. Implicit gender segregation of campus sports spaces (e.g., centralized layout of basketball and soccer courts) may weaken women's sense of belonging to mainstream sports, and thus they face higher psychological costs of participating in physical activity, and their sense of athletic self-efficacy and behavioral intentions to engage in physical activity may be affected. This finding suggests that colleges and universities need to break through the single explanatory framework of physiological differences in their future sports promotion strategies and focus on deconstructing gender barriers in sociocultural constructs, such as developing gender-inclusive physical education and sports curricula and establishing a system of good female role models for sport, to improve the gender differences in self-efficacy for sport and intention to engage in physical activity behaviors.

Details can be found in the revised manuscript on pages 17 to 18, lines 381-404(at the very beginning of the discussion).

Q2: you may need to define the term “good peer relationship” to further contextualize your work. For instance, on number 75 and following numbers, you wrote, 75 when studying issues related to physical activity behaviors [7, 8]. Most studies have found that good 76 peer relationships (?????????????????). It might help to define this concept even if it means putting a synonym in a bracket following it. It is important because this term may be relative. And what is good for a student may be bad for another.

A2: We are very grateful to the reviewers for this comment. Based on the reviewers' comments, we have defined good peer relations to make this concept clearer in this paper. The modifications are as follows

Some studies have found that good peer relationships (i.e., positive peer interactions through emotional support, joint sports participation, and healthy competition) can promote individuals' engagement in physical activity, which is one of the most important motivations for college students' physical and mental health development. Specifically, under the leadership of peers, individuals will generate behavioral intentions to engage in physical activity through social modeling (e.g., observing peers' exercise behaviors), social facilitation (e.g., peers' presence to enhance exercise performance), and attributional need satisfaction (e.g., integrating into the exercise community), thus participating in sport more actively.

Details can be found in the revised manuscript on page 4, lines 73-80 (Modifications are highlighted in blue).

Q3: However, you can further enhance the study by reporting the effects of age, just as you did for gender. This is important because age can be a determining factor in peer relationship and behavioral intention. For instance, some persons may feel reluctant to associate with people outside their age bracket. So, does age play any role in peer relationship in your study? Also, age may be an influencing factor in students’ behavioral intentions to engage in physical activities. For instance, priorities may change due to age (older students may begin to prioritize academic/work obligations compared to the much younger ones).

A3: We are very grateful to the reviewers for this comment, which has been very beneficial. In the results section of the study, it was obtained that there was no significant difference between peer relationships, social support, exercise self-efficacy, and physical activity behavioral intention in the variable of age, probably because the sample of this study was mainly college students, whose age gap was small. We modified it to read as follows:

Finally, this study found no significant difference in age between peer relationships, social support, exercise self-efficacy, and physical activity behavioral intention, probably because the study group was mainly college students, whose age gap was smaller, and individuals in this age group were closer in physiological and psychological development. The social-ecological system theory emphasizes that individual development is influenced by multi-level environmental factors, including micro-systems (e.g., family, school), meso-systems (e.g., community), and macro-systems (e.g., culture, policy). Based on this theory, the behavior and psychological development of individuals are not only influenced by their own factors, but also by the interaction of the surrounding environment. For college students, they are influenced by similar (schooling, socio-cultural, etc.) environments in the same socio-ecological system, and these similar environmental factors may mask the effects of age on peer relationships, social support, exercise self-efficacy, and physical activity behavioral intentions, resulting in a non-significant age effect.

Details can be found in revised manuscript on page 22 to 23, lines 486-497 (Before the conclusion).

Reviewer#2:

Responses to comment: (Revisions are in light blue, for example AAA)

Q1: The sample consists of only 514 students. While this number may be sufficient for certain statistical analyses, it does not adequately capture the diversity of university students, particularly across different cultures or universities with varied educational programs.

A1: We are grateful to the reviewers for this comment, as this study focused on exploring the mechanisms influencing physical activity behavioral intentions in a population of college students, only the demographic variables of gender, school year, and age were considered, and not cross-cultural backgrounds and colleges and universities with different educational systems. In response to this comment, in the section on research subjects, the G*Power test minimum sample size was added to ensure the scientific nature of sample size selection. We modified it to read as follows:

“In this study, before the questionnaire was administered, the minimum sample size was calculated using the G*Power program, setting α=0.05, β=0.80, Effect Size=0.15, to obtain the study's required the minimum sample size was 77”.

In addition, in response to this question from the reviewer, the research team will consider cross-cultural contexts and different educational systems in future studies to explain the relationship more fully.

Details can be found in the revised manuscript on page 10, lines 209-211(This section is in light blue).

Q2�The study is based on theories; however, it does not provide sufficient details on how these theories practically support the hypotheses.

A2: We are very grateful to the reviewers for this comment, which made us realize the lack of support for the theoretical foundation. This study has revised the introduction section based on the suggestion. The revisions are as follows:

Physical activity is an indispensable part of people's lives, as it reflects their sporting values, physical fitness, and mental health. Analyzing from a doctrinal perspective, existing studies have mainly explored the relationships between peer relationships and physical activity behavior, social support and physical activity behavior, and exercise self-efficacy and physical activity behavior. Some studies have found that good peer relationships (i.e., positive peer interactions through emotional support, joint sports participation, and healthy competition) can promote individuals' engagement in physical activity, which is one of the most important motivations for college students' physical and mental health development. Specifically, under the leadership of peers, individuals will generate behavioral intentions to engage in physical activity through social modeling (e.g., observing peers' exercise behaviors), social facilitation (e.g., peers' presence to enhance exercise performance), and attributional need satisfaction (e.g., integrating into the exercise community), thus participating in sport more actively. Although many previous studies have explored the influence mechanism of physical activity and achieved certain results, few scholars have explored the influence mechanism of the antecedent factor of physical activity behavior - the intention of physical activity behavior. So, what is the influence mechanism of physical activity behavior intention? Is it more influenced by intrinsic or extrinsic influences? According to existing research, the formation of the intention to engage in physical activity is a complex psychological process that is influenced by a combination of factors. Bandura's triadic interaction theory suggests that environmental factors (e.g., peer relationships, and social support) form a cyclic dynamic with behavioral intention through personal cognition (e.g., self-efficacy). Based on the ternary interaction theory, this study introduces peer relationship and social support among environmental factors and exercises self-efficacy among personal cognitive factors to explore the influence mechanism of physical activity behavioral intention, aiming at enriching the research results in the existing field and providing theoretical support for the development of physical education and lifelong physical education in colleges and universities.

Details can be found in the revised manuscript on pages 4 to 5, lines 68-92(This section is in light blue).

Q3a: The study did not clarify whether the normality of the data distribution was verified using tests such as Shapiro-Wilk or Kolmogorov-Smirnov, which is essential when employing tests like t-tests or analysis of variance (ANOVA).

A3a: We are very grateful to the reviewers for their comments, which made us aware of the lack of rig

---

## [Decision Letter · Decision Letter 1]

26 Feb 2025

The effect of peer relationships on college students' behavioral intentions to be physically active: the chain-mediated role of social support and exercise self-efficacy

PONE-D-24-44384R1

Dear Dr. zhang,

We’re pleased to inform you that your manuscript has been judged scientifically suitable for publication and will be formally accepted for publication once it meets all outstanding technical requirements.

Kind regards,

Jianpeng Fan

Academic Editor

PLOS ONE

Additional Editor Comments (optional):

Reviewers' comments:

Reviewer's Responses to Questions

**Comments to the Author**

1. If the authors have adequately addressed your comments raised in a previous round of review and you feel that this manuscript is now acceptable for publication, you may indicate that here to bypass the “Comments to the Author” section, enter your conflict of interest statement in the “Confidential to Editor” section, and submit your "Accept" recommendation.

Reviewer #1: All comments have been addressed

Reviewer #2: All comments have been addressed

Reviewer #3: All comments have been addressed

2. Is the manuscript technically sound, and do the data support the conclusions?

Reviewer #1: Yes

Reviewer #2: Yes

Reviewer #3: Yes

3. Has the statistical analysis been performed appropriately and rigorously? 

Reviewer #1: Yes

Reviewer #2: Yes

Reviewer #3: Yes

4. Have the authors made all data underlying the findings in their manuscript fully available?

Reviewer #1: Yes

Reviewer #2: Yes

Reviewer #3: Yes

5. Is the manuscript presented in an intelligible fashion and written in standard English?

Reviewer #1: Yes

Reviewer #2: Yes

Reviewer #3: Yes

6. Review Comments to the Author

Reviewer #1: Having compared my previous review with the edits made by the authors, I am pleased to state that all my comments regarding the manuscript have been adequately addressed. Thank you.

Reviewer #2: All comments have been answered. The researchers have reviewed the required comments and responded to them appropriately in the research.

Reviewer #3: Appreciation is extended to the authors for their endeavors to enhance the quality of the submitted article. My comments and concerns have been addressed.

7. PLOS authors have the option to publish the peer review history of their article (what does this mean? ). If published, this will include your full peer review and any attached files.

**Do you want your identity to be public for this peer review?** For information about this choice, including consent withdrawal, please see our Privacy Policy .

Reviewer #1: No

Reviewer #2: No

Reviewer #3: No

---

## [Editor Report · Acceptance letter]

PONE-D-24-44384R1

PLOS ONE

Dear Dr. zhang,

I'm pleased to inform you that your manuscript has been deemed suitable for publication in PLOS ONE. Congratulations! Your manuscript is now being handed over to our production team.

Kind regards,

on behalf of

Dr. Jianpeng Fan

Academic Editor

PLOS ONE